# Cancer-Homing CAR-T Cells and Endogenous Immune Population Dynamics

**DOI:** 10.3390/ijms23010405

**Published:** 2021-12-30

**Authors:** Emanuela Guerra, Roberta Di Pietro, Mariangela Basile, Marco Trerotola, Saverio Alberti

**Affiliations:** 1Center for Advanced Studies and Technology (CAST), Laboratory of Cancer Pathology, University “G. d’Annunzio”, 66100 Chieti, Italy; emanuela.guerra@unich.it (E.G.); marco.trerotola@unich.it (M.T.); 2Department of Medical, Oral and Biotechnological Sciences, University “G. d’Annunzio”, 66100 Chieti, Italy; 3Department of Medicine and Aging Sciences, Section of Biomorphology, University “G. d’Annunzio”, 66100 Chieti, Italy; r.dipietro@unich.it (R.D.P.); mariangelabasile@hotmail.it (M.B.); 4Unit of Medical Genetics, Department of Biomedical Sciences, University of Messina, 98122 Messina, Italy

**Keywords:** CAR-T cells, immune cell populations, signaling, immune checkpoint blockade, cancer

## Abstract

Chimeric antigen receptor (CAR) therapy is based on patient blood-derived T cells and natural killer cells, which are engineered in vitro to recognize a target antigen in cancer cells. Most CAR-T recognize target antigens through immunoglobulin antigen-binding regions. Hence, CAR-T cells do not require the major histocompatibility complex presentation of a target peptide. CAR-T therapy has been tremendously successful in the treatment of leukemias. On the other hand, the clinical efficacy of CAR-T cells is rarely detected against solid tumors. CAR-T-cell therapy of cancer faces many hurdles, starting from the administration of engineered cells, wherein CAR-T cells must encounter the correct chemotactic signals to traffic to the tumor in sufficient numbers. Additional obstacles arise from the hostile environment that cancers provide to CAR-T cells. Intense efforts have gone into tackling these pitfalls. However, we argue that some CAR-engineering strategies may risk missing the bigger picture, i.e., that a successful CAR-T-cell therapy must efficiently intertwine with the complex and heterogeneous responses that the body has already mounted against the tumor. Recent findings lend support to this model.

## 1. Introduction

Chimeric antigen receptor (CAR) therapy is based on patients’ blood-derived T cells and natural killer (NK) cells, which are engineered in vitro to express artificial receptors to recognize a specific target antigen and the cells expressing that antigen (Figure 1) [1].

The genetic modification of peripherally derived T cells with CAR was developed from the concept of adoptive immunotherapy using tumor-infiltrating lymphocytes (TIL) [2], whose T cell receptors (TCR) can recognize tumor-associated antigens. TCR require the presentation of processed antigens by the major histocompatibility complex (MHC). This led to the suggestion that T cells TCR with high affinity for MHC/peptide, whether of syngeneic or allogeneic origin, may be utilized for efficient anticancer immunotherapy [3].

CAR-T cells were instead developed to recognize the target antigen through immunoglobulin antigen-binding regions. Hence, CAR-T do not require target-peptide presentation by MHC molecules. This correspondingly allowed CAR constructs to be used to direct the activity of NK cells [4].

CAR-T cell therapy has been tremendously successful in the treatment of leukemias [5,6,7,8], seminal to this success being the development of anti-CD19 CAR-T. CAR-T immunotherapy has subsequently undergone intense testing for application to the treatment of solid tumors [9,10]. However, the clinical efficacy of CAR-T cells observed in hematological malignancies is rarely found against solid tumors.

CAR-T cell therapy for solid tumors faces many hurdles [9,10], starting from the very first step of intravenous administration of activated CAR-T cells, wherein in vitro engineered cells must be driven to the tumor by appropriate molecular signals. Correspondingly, early (≤ 3 days) intratumoral infiltration of CAR-T cells post-infusion was shown to be a predictor of survival, and lymphokines/cytokines that increase homing to the tumor were shown to be essential for CAR-T cell therapy outcome [11]. Additional challenges for CAR-T cells are provided by the requirement of long-term persistence within the tumor, resistance to exhaustion, and distinct interaction with other cancer infiltrating cell populations, within the largely immunosuppressive cancer environment [12]. The induction of immunosuppression was assessed using CAR targeting mesothelin or fibroblast activation protein and assessing the functional capacity of CAR-T lymphocytes infiltrating a tumor (CAR-TIL). CAR-TIL underwent rapid loss of functional activity in the tumor. This hypofunction was reversible when the T cells were isolated away from the tumor [13]. Among the factors associated with CAR-T cell hypofunction, there are the upregulation of intrinsic T-cell inhibitory enzymes (diacylglycerol kinase and SHP-1) and expression of surface inhibitory receptors (PD-1-programmed cell death protein-1, LAG3-lymphocyte-activation gene 3, TIM-3-T-cell immunoglobulin and mucin-domain containing-3, and 2B4). This confirms the immunosuppressive impact of tumor environment and shows that CAR-T cell inactivation is reversible, suggesting the feasibility of systemic approaches to overcome this tumor-induced inhibition, which include PD-1 pathway antagonism [13].

Intense efforts have gone into tackling these pitfalls. However, we argue that some CAR-engineering strategies may risk missing the big picture, i.e., that a successful CAR-T-cell therapy must efficiently intertwine with the complex and heterogeneous responses that the body has already mounted against the tumor [14]. Experimental evidence has been recently obtained that lends support to this model.

## 2. CAR Design

### 2.1. Early CAR

The concept of CAR was first proposed by Kuwana et al. in 1987 [15]. A CAR typically comprises an antibody fragment, such as an scFv or Fab fragment, incorporated in a fusion protein that contains additional components, such as a CD3*ζ* and a CD28 transmembrane domain, together with selective T-cell activating moieties, including the endodomains of CD28, OX40, 4-IBB, Lek, and ICOS (Figure 2) [8].

The first-generation CAR construct comprised a CD3*ζ* chain as a key transmitter of signals from endogenous TCR. However, cocd47-stimulation of the TCR is required for the efficient recognition of an MHC–antigen complex in a physiological TCR context. Hence, improvements in the CAR design included secondary and tertiary intracellular signaling chains, that enhanced activatory signaling. These subsequent generations of CAR, with the addition of one versus two co-stimulatory domains (in 2nd and 3rd generation CAR, respectively) have shown enhanced activity, persistence, and efficacy (Figure 3) [12].

First-generation CAR-T entered clinical trials for leukemia, lymphoma, and various other types of cancer, including ovarian cancer and neuroblastoma [1]. Lack of activation resulted in ineffective anti-solid tumor action, but persistent exposure to the tumor environment led to some continued therapeutic effects in patients with B-cell lymphoma and neuroblastoma [1,16].

Second-generation CAR constructs [6] provided a dual signaling function for T-cell activation. The second-generation CAR aimed at integrating intracellular signaling domains from several co-stimulatory molecules, such as CD28, 4-1BB, or CD137, inducible T cell co-stimulator (ICOS) or CD278, OX40, or CD134 fused to the cytoplasmic tail of the CAR, thus amplifying CAR signaling [17]. A large degree of intergenerational variety exists between second- and third-generation CAR, with a range of co-stimulatory domains being tested (CD28, 4-1BB, OX40, CD27, ICOS, DAP10, and LAT-linker for activation of T cells) [12]. The attributes of each co-stimulatory domain differ in their ability to confer cytokine secretion, cytotoxicity, proliferation, and memory development to the modified-CAR-T cells [18].

Third-generation CAR comprised three or more signaling functions, typically incorporating CD28 transmembrane and endodomains, fused to the signaling subunits of 4-1BB, OX40, or Lek, and to the cytoplasmic domain of CD3ζ. A third-generation CAR consisting of αCD19-CD3*ζ*-CD28-4-1BB was shown to lead to complete remissions in patients with chronic lymphocytic leukemia [19].

### 2.2. Fourth-Generation CAR

Co-stimulation of TIL with cytokines such as IL-2 and IL-12 had previously been utilized to enhance anticancer responses [20]. A corresponding benefit was obtained by adding immune checkpoint blockers (ICB) [21]. The vast use of ICB in anticancer therapy in recent years has allowed to unravel distinct mechanistic aspects in adoptive immunotherapy [22] and is now helping to shed light on corresponding issue on CAR-T cell therapy. Consistently, PD-1 was shown to be upregulated on exhausted CAR-T cells and TIL, leading to loss-of-function T lymphocytes. The enhanced efficacy of CAR-T cells was correspondingly obtained by adding ICB to CAR-T cell therapy [23]. Additional insight was provided by exploiting the cancer-immune system liaison using bispecific antibodies [24]. Altogether, these approaches indicated that making use of cancer-infiltrating immune cells, while tilting the balance of immune regulatory circuits toward activation, was required to reach efficacy. Following these principles, to induce a pro-activatory milieu, CAR-T cells were engineered to release transgenic cytokines upon CAR signaling in the targeted tumor tissue. Such ‘T cells redirected for antigen-unrestricted cytokine-initiated killing’ (TRUCK) are also called ‘fourth-generation’ CAR-T cells. Through CAR-induced release, cytokines were secreted in the target tissue rather than in the circulation, thus alleviating systemic side effects. The TRUCK concept is currently being explored using a panel of cytokines, such as IL-7, IL-12, IL-15, IL-18, IL-23 (Figure 4), and combinations thereof [25]. The secretion of IL-15 or IL-18 enhances T cell proliferation. The combination of CCL19 and IL-7 recruits endogenous immune cells and establishes a memory response against tumor cells [26]. CAR-T cells engineered to secrete IL-12 have been shown to remodel the tumor microenvironment (TME) by reprogramming tumor-associated macrophages (TAM) to an M1 phenotype and decreasing the presence of myeloid-derived suppressor cells (MDSC) and Treg in syngeneic mouse models [27]. Similarly, CAR-T cells that constitutively secrete IL-18 increase intratumoral M1 macrophages, activated dendritic cells (DC), and activated NK cells numbers, while decreasing M2 macrophages and Treg (regulatory T cells) levels. A direct comparison of CAR-T cells expressing IL-12 to IL-18 indicated that IL-18 is more effective at remodeling the immunosuppressive TME in a syngeneic murine pancreatic cancer model [28].

CAR-T cells engineered to secrete interleukin(IL)-12 stimulate innate immune cells against the tumor and become resistant to Treg and MDSC [29]. T cells were engineered with the IL-23 p40 receptor chain that associates with the endogenous IL23a p19 receptor to use released IL-23 in an autocrine fashion. CAR-T cells with engineered IL-23 receptor showed superior reactivity in various tumor models and attenuated side effects, suggesting an advantageous use of IL-23 for sustaining an improved CAR-T cell response [30].

TRUCK approaches are being tested in early phase clinical trials. The NCT03182816 trial was started in 2017 as a phase I/II single-arm to determine the safety and efficacy of infusion of autologous T cells engineered to express immune checkpoint antibodies (CTLA-4 and PD-1) and chimeric antigen receptor targeting epidermal growth factor receptor (EGFR-CAR) in adult patients with EGFR-positive advanced malignant solid tumors. Eight of nine patients showed detectable EGFR-CAR-T cells in their peripheral blood. One patient had a partial response, which lasted >13 months, six had stable disease, and two had disease progression [31]. Koneru et al. assessed in a phase I clinical trial IL-12-secreting CAR-T cells directed against MUC-16(ecto) for recurrent ovarian cancer [32]. The EGFR-IL-12-CAR-T NCT03542799 clinical trial started enrolling patients with metastatic colorectal cancer in 2018. This phase I/II study is expected to accrue 20 patients by 2021. The NCT03932565 phase I clinical trial is assessing CAR-T cells directed against Nectin4_FAP since 2019. CAR-T cells were engineered to express IL-7 and CCL19 (C-C Motif Chemokine Ligand 19) [26] or IL-12. The accrual of 30 patients is expected to be completed by 2021.

## 3. Main Challenges for CAR-T Cell Therapy against Solid Tumors

As mentioned above, the clinical efficacy of CAR-T cells observed in hematological malignancies is rarely found against solid tumors. Hematological malignancies are devoid of many of the physical immunosuppressive factors that hamper adoptively transferred cells from reaching solid tumors. Furthermore, target antigens that are present on hematological cancers are often homogenous and are expressed in most tumor cells. In contrast, target antigens on solid tumors are often heterogeneous both across tumors and between primary and metastatic sites. CAR-T cell therapy for solid tumors faces additional obstacles [9,10], starting from the administration route, wherein CAR-T cells injected in the peripheral blood must encounter the correct chemotactic signals to traffic to the tumor in sufficient numbers. The abnormal cancer vasculature impedes efficient passage (diapedesis) toward tumor tissues [33,34,35]. Physical barriers by hard-to-penetrate fibrotic stroma prevent adequate CAR-T cell diffusion at a distance from the blood vessels. Finally, immunosuppressive factors include inhibitory checkpoint pathway signals and immunosuppressive cytokines [36], TGF(transforming growth factor)-β among them [37,38,39]. Products of an altered metabolism, including amino acids, lipids, and DNA base precursors [40,41,42], and reactive oxygen species (ROS) [43] can be found in the tumor environment [12]. All these factors considerably affect the long-term persistence within the tumor in an active state.

### 3.1. CAR Target Choice for Anticancer Therapy

An appropriate choice of target antigens and the effective combination with immunostimulatory signaling have been shown to enhance CAR efficacy. CAR comprising the ICOS signaling domain liaises with the effective antitumor effect on gliomas that express the epidermal growth factor receptor variant III (EGFRvIII) [44]. The preclinical evaluation of CAR-T cell therapy targeting the tumor antigen 5T4 in ovarian cancer led to successful outcomes [45]. CAR targets included genetic products arising from gene mutations (*EGFRvIII*) [46], modified glycosylation patterns (MUC1) [47], cancer-testis antigen-derived peptides (MAGE), and mesothelin-specific CAR-T cells [48]. CAR was further generated that target overexpressed antigens in breast cancer, lung cancer, and pancreatic cancer, such as carcinoembryonic antigen GD2, prostate-specific membrane antigen, HER2/ERBB2, MUC16 [49] or tumor stroma (fibroblast activation protein or vascular endothelial growth factor receptor, VEGFR) [50]. Additional novel targets include the type 1 insulin-like growth factor receptor (IGF1R) and receptor tyrosine kinase-like orphan receptor 1 (ROR1) for sarcoma as well as the L1-cell adhesion molecule (L1-CAM) for ovarian cancer [51].

These combined efforts have led to a burgeoning of clinical trials in the field, the majority remaining focused, though, on leukemias and lymphomas [12]. FDA-approved CAR-T cell therapies now include CTL019 (Kymriah) [52], KTE-C19 (Yescarta) [53], JCAR017 (Breyanzi) [54], KTE-X19 (Tecartus) [55], and bb2121 (Abecma) [56].

### 3.2. Trafficking and Persistence of CAR-T Cells in the Tumor

The persistence, trafficking, and maintenance of function remain a challenge in many CAR-T approaches. Using intravital imaging, anti-CD19 CAR-T cells were tracked at lymphoma sites [57]. However, circulating targets trapped CAR-T cells in the lungs, reducing their access to lymphoid organs. In the bone marrow, tumor apoptosis was largely due to CAR-T cells that engaged, killed, and then detached from their targets within 25 min. Notably, not all CAR-T cell contacts elicited calcium signaling or killing while interacting with tumors, uncovering extensive functional heterogeneity. Mathematical modeling revealed that direct killing was sufficient for tumor regression. Finally, antigen-loss variants were shown to emerge in the bone marrow, but not in lymph nodes, where CAR-T cell cytotoxic activity was reduced. Hence, a previously unappreciated level of diversity exists in the outcomes of CAR-T cell interactions in distinct anatomical districts in the body, with important clinical implications [57].

### 3.3. CAR-T Treatment Toxicity

The anticancer activity of CAR-T can associate with life-threatening toxicity due to the increased secretion of pro-inflammatory cytokines (cytokine release syndrome, CRS) and the immune effector cell-associated neurotoxicity syndrome (ICANS) as mediated by the simultaneous activation of large numbers of T cells [8,58]. This frequently leads to multiorgan dysfunction, pulmonary failure, and death [59]. Consequent therapeutic intervention requires the intense management of treated patients [60,61].

Cytokine release is usually greater with CAR containing CD28 versus 4-1BB co-stimulatory domains. On the other hand, constructs with either domain confer similar anticancer activity in mouse models. T cell products expressing CAR with either CD28 or 4-1BB co-stimulatory domains have been highly efficacious in patients with relapsed hematological malignancies, and anti-CD19 CAR showed similar activity regardless of the source of the co-stimulatory domain. In large-cohort clinical trials, the rates of neurological toxicities have been higher with CD28–co-stimulated CAR, although this finding is probably the result of multiple converging factors rather than due to CD28 signaling alone [62]. One of these factors was the increased circulating IL-17 levels at baseline in patients with locoregional metastatic melanoma. In ICB approaches, increased IL-6 levels in patients with metastatic melanoma treated with ipilimumab (anti-CTLA-4 antibody) were shown to be associated with more intense adverse events. More global cytokine dysregulation as assessed by measuring the circulating levels of several cytokines at baseline or early on treatment has been shown to be predictive of adverse events in patients treated with anti-PD-1 therapies alone or in combination with anti-CTLA-4 therapies [21].

Intense cytokine release is associated with T-cell activation upon engagement with target cells, which leads to higher levels of circulating cytokines, including IL-6 and interferon γ. Consistently, elevated levels of serum cytokines are less common in patients who do not have a clinical response after CAR-T cell therapy [63].

NK cells produce several cytokines, including tumor necrosis factor α, interferon γ, and IL-10 [64]. NK cells produce lower levels of IL-6 than T cells [63], thus potentially reducing systemic cytokine-storm toxicity. Of note, the CAR-T cell-induced cytokine release syndrome can be mediated by macrophages and is relieved by IL-1 blockade [65].

### 3.4. CAR-T Cell Exhaustion

A major concern of CAR-T therapy is that CAR-T cells may become exhausted or dysfunctional [66,67], which is a phenomenon whereby the T cells become unresponsive due to overstimulation [68].

CAR-T cell exhaustion is considered the outcome of the chronic tumor stimulation imposed on T cells, which leads to disruption of their function [67]. Proof of (re-)activation of TIL upon blockade of the PD-1/PD-L1 (programmed death-ligand 1) or other checkpoint axes [69] has strongly supported this model. Correspondingly, other therapeutic measures, including adoptive T-cell therapy, epigenetic reprogramming, antibodies targeting T-cell co-stimulatory molecules, metabolic reprogramming, and conventional cancer therapies, such as chemotherapy, radiotherapy, and targeted therapy, have been implemented to enhance antitumor immunity, overcome resistance, and increase therapeutic efficacy [22].

Cell exhaustion may differentially affect T cell subpopulations, suggesting a relationship with the selection procedures utilized to manufacture CAR-T cells. Optimal cell subpopulations for adoptive cell transfer were suggested to be those that retain their memory/naïve capacities [70] to permit a greater boost in proliferation and function in vivo. Wnt signaling was shown to promote the generation of CD44(low)CD62L(high)Sca-1(high)CD122(high)Bcl-2(high) self-renewing multipotent CD8(+) memory stem cells with proliferative and antitumor capacities exceeding those of central and effector memory T cell subsets [70].

Epigenetic profiles regulate the gene expression of key transcription factors over immune cell differentiation and proliferation pathways. Through a screening of chemical probes with defined epigenetic targets, JQ1, an inhibitor of bromodomain and extra-terminal motif (BET) proteins, was found to maintain CD8+ T cells with functional properties of stem cell–like and central memory T cells [71]. Adoptively transferred in vitro-JQ1-treated CAR-T cells showed higher proliferation, persistence, and increased cytokine secretion than non-treated CAR-T cells in murine models [71].

The altered differentiation of CAR-T cells can also accompany T cell exhaustion [72]. Utilizing mesothelin-redirected CAR-T cells in pancreatic cancer, CAR dysregulation was found to be associated with a CD8+ T-to-NK-like T cell transition, as driven by SOX4 (SRY-Box Transcription Factor 4) and ID3. The downmodulation of ID3 and SOX4 expression was indicated to improve the efficacy of CAR-T cells in solid tumors by preventing or delaying T cell exhaustion [72]. These findings further reveal CAR-T cells’ plasticity as a main actor of immune response dynamics.

### 3.5. Counteracting Immunosuppression against CAR-T Cells

A major hurdle to be overcome by tumor-infiltrating CAR-T cells is immunosuppression. T cell exhaustion itself was shown to be an outcome of cancer-associated immunosuppression. Correspondingly, resistance to exhaustion was shown to be linked to interaction with other cancer-infiltrating cell types [12].

Cancers contain a broad ensemble of genetically normal cells within an extracellular matrix (ECM), which was collectively termed the tumor microenvironment (TME), that substantially diverges from normal stroma. This has fostered the concept that tumors grow as integrated tissues or organs, that combine diverse components, such as vasculature, nerves, an immune environment, and connective tissue.

TGF-β is a key driver of immunosuppression [37,38,39]. Prostate cancer, in particular, secretes TGF-β as a means to inhibit immunity while allowing for cancer progression. Blocking TGF-β signaling augments T cell ability to infiltrate, proliferate, and mediate antitumor responses in prostate cancer models. The potency of PSMA(prostate-specific membrane antigen)-targeted CAR-T cells was correspondingly enhanced utilizing dominant-negative TGF-βRII (dnTGF-*β*RII) expression in CAR-T cells [73]. This led to an increased proliferation of CAR-T cells, enhanced cytokine secretion, resistance to exhaustion, long-term in vivo persistence, and eradication of human prostate cancer in mouse models. A phase I clinical trial is being conducted to assess these CAR-T cells in relapsed and refractory metastatic prostate cancer patients. This is a non-randomized, sequential assignment, open label trial; 18 participants have been enrolled; completion of the study is expected in 2022 (ClinicalTrials.gov: NCT03089203) [73]. Knocking out the endogenous TGF-β receptor II (TGFBR2) in CAR-T cells was achieved using CRISPR/Cas9. This reduced Treg conversion prevented CAR-T cell exhaustion and achieved higher tumor eradication rates both for xenografts and for PDX (patient derived xenograft), with higher proportion of memory and effector memory CAR-T cell subsets [74]. Additional approaches were suggested to be effective, e.g., the expression of chimeric TGFBR2 and TGFBR1 where the TGF-β-binding domain is fused to the transmembrane and intracellular signaling domains of IL-12 receptor (CTBR). CAR-T/CTBR cells secreted significantly greater amounts of IFNγ than control T cells following activation in the presence of TGF-β. In the absence of IL-2, antigen-driven expansion was severely limited by exposure to TGF-β, and CAR-T cells progressively lost cytotoxic activity. Although T cells overexpressing the dominant negative TGF-β receptor failed to expand and clear tumor cells in the presence of TGF-β, CTBR expressing CAR-T cells maintained their ability to expand and kill tumor targets in the presence of TGF-β [75].

Metabolic conditions can negatively impact on CAR-T cell function: among them, an increase in the acidity of the TME because of increased glycolysis by cancer cells. This ‘Warburg effect’ [76,77] stems from a preferential utilization of glucose via glycolysis rather than via oxidative phosphorylation. Cancer-associated fibroblasts (CAF) contribute to increased intratumor glycolysis and impact on breast cancer growth [78]. High glycolysis leads also to an increase in oxidative stress and in ROS production. The secretion of high levels of ROS by MDSC contributes additional immunosuppressive capacity. Immunosuppression by oxidative stress impairs CAR-T cells proliferation and cytotoxicity [12]. This led to engineering CAR-T cells to secrete catalase (CAT), an antioxidant enzyme, into the TME. CAR-CAT-T cells were shown to regain their antitumor functions [43]. Local catalase secretion provided a bystander effect and restored cytotoxic function to NK cells.

One of the approaches used to fight immunosuppression has been the generation of CAR-T cells expressing cell-surface dominant-negative receptors (DNR) to override the inactivating signals present in the TME. DNR can be generated with a functional extra-cellular domain and a mutation in the intracellular region to abolish downstream signal transduction. DNR can effectively compete with their endogenous counterparts. The use of DNR for immunosuppressive factors such as TGF-β has endowed transduced EBV cells with resistance to immunosuppression [79]. A DNR for PD-1 on CAR-T cells rescued the effect of checkpoint blockade and restored effector functions. PD-1/PD-L1 blockade is normally achieved through systemic antibody delivery, which can result in autoimmune reactions. PD-1 ‘insensitive’ DNR T cells do not require systemic ICB and may prevent this major side effect. Switch receptors offer yet another approach to circumvent immunosuppression. These CAR contain the extracellular portion of an antibody specific for an immunosuppressive molecule, such as PD-1 or CTLA-4, which is fused to an intracellular activating signaling molecule, such as CD28. The infiltration and antitumor efficacy of PD-1-CD28 switch-CAR-T cells were enhanced versus parental CAR-T cells [80]. Switch-CAR-T cells showed a reduction in other checkpoint inhibitors, e.g., LAG3, TIM-3, and CEACAM1 (carcinoembryonic antigen-related cell adhesion molecule 1), and increased IL-2 signaling, suggesting an induction of recovery from cell exhaustion.

The reduction of inhibitory signaling pathways in T cells has shown promise in T-cell reactivation. The inhibition of Protein Kinase A with Ezrin using a ‘regulatory subunit 1 anchoring disruptor’ (RIAD-CAR) resulted in an upregulation of CXCR3 and CD49D integrin (VLA-4), which resulted in enhanced RIAD-CAR-T cells trafficking to tumors and better migration to CXCL10 in vitro [81]. RIAD-CAR cells expressed higher levels of both IFN*γ* and cytotoxicity and were more resistant to immunosuppression in TME.

IL-8 release within tumors was utilized to enhance intratumoral T-cell trafficking. Modified CAR inducing the expression of IL-8 receptors, CXCR1 or CXCR2, showed enhanced migration and persistence of T cells in the tumor. This induced complete tumor regression and long-lasting immunologic memory in preclinical models of glioblastoma, ovarian, and pancreatic cancer [11].

TGF-β-resistant EGFRvIII CAR-T cells were shown to possess higher antitumor efficacy in murine glioma models [82], and prolonged survival was observed following EGFRvIII CAR-T cell treatment in advanced glioblastoma [83].

### 3.6. CAR-T Cells Targeting Multiple Antigens

The approaches described above supported the feasibility of ‘multi-targeted CAR-T’. The rationale was to improve selectivity for cancer cells and to reduce the off-tumor effects based on the presence or absence of two target antigens. This was also expected to lead to enhanced cytotoxicity and to reduce chances of antigen escape variants. Early work showed the synergistic effects of two individual CAR against two separate targets, i.e., folate binding protein and Her-2, together with limited antitumor efficacy in the presence of only one [84]. Dual CAR-expressing T cells were also shown to secrete higher cytokine levels than single CAR-T cells when co-cultured with dual antigen-expressing tumor targets [84].

Distinct intracellular signaling domains were also engineered onto different CAR recognizing two separate target antigens [85], both providing suboptimal activation upon binding of their target antigen. The rationale was to generate CAR-T cells that were only able to function at full capacity in the presence of both target antigens, thus limiting activation if only a single antigen was present. This concept was tested by targeting the prostate tumor antigens PSMA and PSCA (prostate stem cell antigen). Co-transduced T cells were shown to kill tumors that expressed both antigens but not tumors expressing either antigen alone [85].

Dual recognition was also exploited to direct CAR-T cells to the tumor target (CAR one) and then kill cancer cells (CAR two). One of these approaches utilized a Synthetic Notch receptor (SynNotch) with an extracellular CAR specific for a target antigen, which was fused to an intracellular cleavable transcriptional domain. Binding to the target antigen resulted in Notch cleavage and the downstream transcriptional activation of Notch-inducible genes [86]. Then, a synthetic Notch receptor for one antigen was utilized to induce the expression of a CAR for a second antigen. These dual-receptor CAR-T cells were only activated in the presence of dual-antigen tumor cells [87].

Dual CAR design can also be exploited to enhance signaling by 4-1BB and CD28 domains, which is best achieved by the membrane proximal positioning of both signaling units, within separate, parallel CAR [88].

## 4. Cancer-Associated Immune Cell Populations

Artificial intelligence scrutiny of bulk, single-cell, and spatially resolved gene expression data of CAR-T targets [89] have provided a large-scale identification and validation of cell states and multicellular communities from multiple cancer types. By analyzing these patterns, distinct multicellular communities were discovered that showed unexpectedly strong conservation. These were associated with myeloid and stromal [90] elements, and they were linked to adverse survival, showing that cellular organization in human carcinomas is pivotal for tumor progression and resistance to therapy [91].

### 4.1. T Cell Subsets

T cells mediate antitumor immune responses and are the key target of immune checkpoint therapy, but they can also promote immune tolerance. A clear understanding of the specific contributions and biology of different T cells subsets is required to fully harness the curative potential of immunotherapies [92]. Treg are induced during adoptive anticancer immunotherapy [21,93] and CAR-T cell differentiation [9]. Treg were found to attenuate the cytolytic activity and proliferation of CD8+ T cells during immunotherapy with a bispecific antibody (blinatumomab) [94]. Both CD4+ [95] and CD9+ can exert anticancer activity. Inhibition of CD4+ and CD8+ T cells by Treg allows maintaining self-tolerance [96] through the secretion of immunosuppressive cytokines, e.g., IL-2, IL-10, IL-35, TGF-β, and the expression of checkpoint molecules such as CTLA-4 and PD-1. As identified by the expression of CD4, CD25, and the FoxP3 transcription factor, Treg are frequently found in the tumor microenvironment and in the circulation [21]. The critical role of Treg in the regulation of tumor immunity was validated by preclinical studies where the depletion of Treg in a variety of tumor types could evoke an antitumor immune response. Thus, lymphodepletion may be useful before CAR-T cell therapy as it decreases the number of immunosuppressive Treg as well as competition from other lymphocyte populations [21]. This may allow creating a more favorable environment for CAR-T cells and longer CAR-T cell persistence and efficacy [97]. This may favor CAR-T cells acquisition of memory cell’s function [98,99], which may provide long-term immune recognition of cancer [100] (Figure 5).

### 4.2. Myeloid-Derived Suppressor Cells (MDSC)

MDSC constitute a subset of immune cells that can have immunosuppressive activity in the TME [21,69]. Through different mediators such as arginase 1, inducible nitric oxide synthase, reactive oxygen species, and peroxynitrite, MDSC attenuate the activity of Teff and NK cells, regulate the differentiation of Treg and Breg [102] through the induction of indoleamine 2,3-dioxygenase 1 [103], and induce an immunosuppressive phenotype in macrophages. Moreover, tumor-infiltrating MDSC exhibit a high expression of PD-L1 in the context of various cancer types including colon, ovarian, and bladder cancer. Consistently, MDSC were shown to be targeted and inhibited/depleted by ICB, resulting in an increased Teff to MDSC ratio, which associates to immune response versus resistance [104,105].

The modulation of MDSC function in cancers has achieved remarkable progress. Neutralizing antibody to KIT significantly reduced MDSC expansion and relieved T cell immunosuppression in colon cancer [106]. Antagonists of CXCR2 (S-265610) and CXCR4 (AMD3100) altered the recruitment of immature myeloid cells to the tumor [107]. Anti-IL-6R mAb abrogated the accumulation of MDSC at tumor sites with a subsequent upregulation of IFN*γ* and enhancement of antitumor T-cell responses [108].

### 4.3. Tumor-Associated Macrophages

Tumor-associated macrophages (TAM) play a key role in the regulation of tumor immunity [21,69]. The complex plasticity of TAM parallels a spectrum of phenotypes, with M1 and M2 residing at the two ends of the spectrum. M1 macrophages classically express pro-inflammatory cytokines and promote an antitumor immune response. On the other hand, M2 macrophages are characterized by the expression of anti-inflammatory cytokines and chemokines and suppress CD8+ T cell activation, promote the recruitment of Treg, and contribute to resistance to antitumor immune responses. Tumor cells can escape macrophage-mediated phagocytosis through the expression of anti-phagocytic signals, such as the upregulation of CD47. CD47 binds to SIRPa on macrophages and inhibits macrophage phagocytosis [109]. Thus, anti-CD47 antibodies increase the phagocytosis of cancer cells, representing an efficient strategy of TAM reprogramming [110]. TGF-β1 secretion by macrophages further contributes to an immunosuppressive microenvironment in ovarian cancer [111]. PI3K was shown to be a pivotal switch in TAM function [112]. The expression of ICB such as PD-L1 on these cells further enhances their immunosuppressive effects. Inhibiting the activity of M2-like TAM and redirecting the polarization of macrophages toward the M1 phenotype can enhance response to immunotherapy [113]. A low ratio of adaptive immune response to pro-tumorigenic inflammatory gene signatures in phagocytic myeloid cells is another factor shown to be associated with resistance to PD-L1 blockade in urothelial cancer.

A subset of TAM expresses folate receptor β (FRβ) and possesses an immunosuppressive M2-like profile. Anti-FRβ CAR-T cells targeting TAM achieved the selective elimination of FRβ+ TAM in the TME, resulting in an enrichment of pro-inflammatory monocytes, influx of endogenous tumor-specific CD8+ T cells, delayed tumor progression, and prolonged survival in murine models of ovarian cancer, colon cancer, and melanoma [114].

Advanced approaches of TAM depletion rely on the inhibition of colony-stimulating factor 1 and its receptor (CSF1/CSF1R) signaling [101,115]. Various small molecules inhibiting CSF1R tyrosine kinase have been investigated. Among them, ARRY-382, PLX7486, BLZ945, JNJ-40346527, MCS110, and PD-0360324, which target the intracellular tyrosine kinase of CSF1R, have undergone testing in solid tumors and Hodgkin lymphoma [116]. Targeting TAM with anti-CSF-1R antibody showed effectiveness as anticancer therapy [117].

Bisphosphonates significantly reduced TAM infiltration in lung cancer, melanoma, hepatocellular carcinoma, and lung metastasis from breast cancer [101]. A direct enhancement of metastatic growth was shown by TAM through their effects on breast cancer cell extravasation, survival, and growth. Consistent, liposome-encapsulated clodronate (dichloromethylene diphosphonate) showed effectiveness at both depleting TAM and reducing metastatic burden [118].

The blockade of monocyte recruitment to tumors was utilized as an alternative approach to reduce TAM infiltration. Inhibitors of the chemokine CCL2 or of CCR2, i.e., the receptor for CCL2, have been utilized to this end. CCR2 antagonists inhibit monocyte recruitment and TAM M2 polarization in hepatocellular carcinoma [119]. PF-04136309, a small molecule inhibitor of CCR2, was investigated in a clinical trial of pancreatic cancer, and it was shown to enhance the impact of FOLFIRINOX chemotherapy [120]. Carlumab is a CCL2-targeted antibody, and it successfully represses macrophage infiltration and tumor growth [121]. However, anti-CCL2/CCR2 therapy can have remarkable rebound effects when completed, inducing a ‘flare’ of metastases and accelerating death. This is due to monocyte release from the bone marrow and cancer cell mobilization from the primary tumor as well as blood vessel formation and increased proliferation of metastatic cells in the lungs [122].

IL-10 is a TAM-derived cytokine with the ability to block IL-12 and suppress T-cell tumoricidal function [123]. Inhibition of IL-10 was correspondingly shown to improve the efficacy of chemotherapy [123].

Toll-like receptors (TLR) play a critical role in innate immune response and polarize macrophages into a pro-inflammatory subtype. Agonists of TLR7, TLR8, and TLR9 were shown to revert macrophage polarization and increase tumoricidal activity in several cancer models [124].

Broader approaches to TAM reprogramming include inhibition of DICER, a pivotal enzyme for microRNA synthesis [125], and inhibitors of histone deacetylase [126]. Metabolic interventions can also be utilized for the functional reprogramming of TAM. The reduction of glycolysis was shown to relieve TAM-driven immunosuppression [127]. Lactic acid induces M2-like polarization of TAM, suggesting the inhibition of the production of lactic acid as a viable means to reprogram TAM [128].

Nevertheless, caution must be exerted, as TAM reprogramming is hampered by acquired resistance and compensation by alternative immunosuppressive circuits, and it bears serious side effects, including anemia and autoimmunity [101].

### 4.4. Tumor-Reactive B Lymphocytes

Recent studies have demonstrated a role for B cells in antitumor immunity. The presence of B cells in tumor was associated with a better response to neoadjuvant therapy with ICB in melanoma and renal cell carcinoma. B cells were found primarily in tertiary lymphoid structures (TLS). Tumor-infiltrating B cell populations in responder tumors were enriched in memory B cells. In contrast, naive B cells were more prominent in non-responder tumors. Similarly, in soft-tissue sarcomas, the presence of TLS enriched in B cells was associated with a better response to PD-1 blockade. The mechanism underlying the effect of B cells on response to adoptive immunotherapy is poorly understood. However, mechanisms include the activation of T cells through antigen presentation by memory B cells and B cells-derived cytokines, as well as potential contribution through the production of anticancer antibodies. Future studies are required to determine the precise mechanism of action for these cells as well as the different components of TLS [129,130,131].

### 4.5. NK Cells, Neutrophils, Dendritic Cells (DC)

Other innate immune cells infiltrating the tumor microenvironment, such as NK cells, neutrophils, and DC, can further impact antitumor immune responses. NK cells are cytotoxic lymphocytes that identify target cells in the absence of MHC. NK cells can elicit rapid immune responses [132], and they are essential for natural immunity. NK cell-mediated tumor cell lysis involves distinct receptors, including NKp44, NKp46, NKG2D, NKp30, and DNAM [133]. The activation of NK cells leads to the recruitment and activation of other effectors cell subsets, such as dendritic cells, macrophages, and neutrophils, and to the subsequent activation of antigen-specific T and B cell responses. Tumor-infiltrated neutrophils have shown both pro- and antitumor phenotypes [134,135]. The activities of tumor-associated DC depend significantly on the DC subtype. The tumor microenvironment often dictates an immature phenotype in DC, which are not effective in activating T cells through antigen presentation and promote an immunosuppressive microenvironment through Treg expansion. In contrast, conventional type I DC can effectively stimulate CD8+ T cells in tumor-draining lymph node and within the tumor, creating a novel rationale for therapeutic efforts to increase these cells in the tumor microenvironment. Of relevance, DC targeting, using bispecific T-cell engagers, has been proposed as a general means of T-cell rejuvenation for more durable cancer immunotherapy [102].

## 5. Cancer-Associated Fibroblasts (CAF)

Within the TME, cancer-associated fibroblasts (CAF) assume the role of organizers and direct the recruitment and differentiation of TME-infiltrating cells [136]. These CAF functions were shown to affect tumor growth, maintenance, and progression, and they have an impact on disease outcome [137].

ECM-producing CAF (myofibroblastic, myCAF) are characterized by the expression of PDPN (podoplanin), FAP, ACTA2 (actin α2), COL1A1 (collagent type I α1 chain), and COL1A2. CAF, in addition to ECM-crosslinking enzymes and ECM components, produce ECM-degrading proteases, such as matrix metalloproteinases, for ECM remodeling [138,139]. Vascular CAF (vCAF) is a CAF subset that closely interacts with blood vessels. myCAF are in proximity to cancer cells, whereas inflammatory CAF (iCAF) are more distantly located, suggesting a juxtacrine role of myCAF and a paracrine role for iCAF vs. cancer cells [140]. Such interactions with transformed cells and other cell types change dramatically as cancers evolve, with tumor-promoting and tumor-restraining activities by CAF subsets [141]. Correspondingly, transcriptional profiles of CAF subsets were shown to shift from an immunoregulatory program to wound healing and antigen-presentation programs, indicating CAF function evolution during tumor progression [142]. CAF subsets and their gene signatures were correspondingly shown to vary across cancer types.

CAF can exert broadly divergent functions on tumor cells [143]. CAF with tumor-promoting functions secrete growth factors (HGF, IGF1, CTGF, PDGF, VEGF, LIF), cytokines (IL-1, IL-4, IL-6, IL-8, IL-10, TGF-β), and chemokines (CCL2, CCL5, CXCL5, CXCL9, CXCL10, CXCL12), WNT5α, and bone morphogenic protein 4 (BMP4), which promote tumor progression [144]. Interleukin-10 receptor signaling promotes the maintenance of a PD-1/TCF-1(T cell factor-1)/CD8+/T cell population that sustains antitumor immunity [145]. CD10 and GPR77 (G protein-coupled receptor 77) identify a CAF subset correlated with chemoresistance and poor survival in breast and lung cancer patients by providing a survival niche for cancer stem cells. Notably, targeting these CAF with a neutralizing anti-GPR77 antibody abolishes tumor formation and restores tumor chemosensitivity [146].

vCAF can induce tumor progression via the secretion of CXCL12 and VEGF, which promote angiogenesis. However, CXCL12 and CXCL5 inhibit T cell migration into the tumor [12], and the array of chemokine receptors present on T cells may not match the tumor chemokine signatures, resulting in poor migration to the tumor site [147]. T cells genetically modified to express CXCR2 were shown to migrate toward a host of tumor cells expressing CXCL1. This effect was also observed in mesothelioma and neuroblastoma xenografts using CAR-T cells bearing a CCR2b receptor and in Hodgkin’s lymphoma with CCR4-bearing CAR-T cells [12]. LRRC15 (leucine rich repeat containing 15) is a biomarker of myCAF in pancreatic and breast cancer. The high expression of LRRC15 in myCAF correlated with a lack of response to immunotherapy [148]. myCAF are located at the invasive front of human breast tumors. Inflammatory CAF (iCAF, marked by the expression of CXCL12) are associated with a higher number of TIL. myCAF subset analysis by single-cell RNA sequencing revealed subclusters, which were correlated with the presence of CD4+ T cells expressing PD-1 and/or CTLA4 [36]. An immunosuppressive role of CAF-derived TGF-β modulates the functions of T cells, macrophages, and neutrophils, and increased TGF-β favors immune evasion with T-cell exclusion, preventing acquisition of the T helper 1-effector phenotype [149]. TGF-β signaling was associated with dysregulation of the ECM protein-encoding genes that correlated with higher CD8+ T cells and M1:M2 macrophage ratio [21]. Conversely, TGF-β signaling promotes tumor development by the remodeling of ECM composition and structure [150]. FAP (fibroblast activation protein)-expressing CAF exert an immunosuppressive function [93]. Hence, immunotherapies were designed for the depletion of these CAF or targeting of CAF-derived factors such as CXCL12 or CCL2. Adoptively transferred CAR-T cells targeting FAP selectively reduced FAPhi stromal cells and inhibited the growth of multiple types of transplanted tumors in experimental models. A major mechanism of action of the FAP-CAR-T cells was the augmentation of the endogenous CD8+ T cell antitumor responses [151]. However, this strategy was found to cause significant off-target bone marrow toxicity and cachexia.

CAF can actively influence cancer cells and immune cells through the production of various metabolites, including alanine [40], aspartate, proline [41], lactate, deoxycytidine, and lipids (such as lysophosphatidic acid) [42]. A more immunosuppressive stroma is characterized by the upregulation of CXCL5 in cancer cells, which results in increased recruitment of MDSC and suppression of CD8+ T cells [36].

CAF with tumor-restraining functions promote antitumor immunity, generate a pro-inflammatory secretome, provide tumor-inhibitory signaling, and produce ECM components, which enhance intra- and peri-tumoral fibrosis, and correspondingly reduce tumor cell invasion and dissemination capacity. Higher tumoral αSMA levels and stromal densities are associated with higher overall survival in patients with pancreatic cancer [152,153]. Targeting the Sonic hedgehog (SHH)–Smoothened (SMO) signaling axis increases cancer cell proliferation, owing to inhibition of the tumor-restraining phenotypes of myCAF by SHH-SMO [154]. Tumor-restraining functions of SHH–SMO signaling, and the fibrotic tumor stroma have also been identified in the context of multiple cancer types. A subpopulation of antigen-presenting CAF (apCAF) is characterized by MHC class II and CD74 expression in pancreatic tumors and other cancer types [141]. However, apCAF lack classical co-stimulatory molecules expressed by professional antigen-presenting cells (APC). Nevertheless, the engineering of designer APC by reconstituting elements of antigen presentation in fibroblastoid cells showed that this allows functional interactions with the immune system [155,156].

## 6. Immune Cell Dynamics and CAR-T Anticancer Response

As suggested above, increasing experimental evidence indicates that a successful CAR-T-cell therapy must efficiently intertwine with the complex and heterogeneous responses that the body has already mounted against the tumor [14]. A robust systemic immune response was shown to be essential to the success of cancer immunotherapies [93] together with the modulation of existing systemic immunity [157] and of the host metabolism [158]. Various types of innate and adaptive immune cells reside within or infiltrate the tumor microenvironment, and the dynamic crosstalk between these immune cells and tumor cells defines the immune status of the tumor and can promote or hinder antitumor immunity.

### 6.1. CAR-T Cell Interactions with the Tumor Micro-Environment

Cellular and humoral responses are highly dynamic in response to infection and cancer [159]. Consistent, adoptive cell therapy with tumor-specific Th9 cells induces viral mimicry to eliminate antigen-loss-variant tumor cells [160] and get rid of infiltrated tumors [161].

As tumors evolve, the tumor microenvironment gradually becomes more immunosuppressive with several components of the innate and adaptive immune system contributing to tumor immune evasion and to resistance to immunotherapeutic tools. A key player in immunosuppression is TGF-β, and TGF-β-counteracting therapies have shown favorable impact in distinct settings [79]. However, TGF-β1 operates as an activation signal for Vγ9Vδ2 T cells in the adoptive immunotherapy of cancer [162], indicating the need for a careful assessment of therapeutic interventions to enhance the synergy of diverse tumor-infiltrating immune cell populations in vivo. An intriguing consistent strategy is that of switching TGF-β from an immunosuppressive cytokine into a T cell stimulant. A CAR was designed that responds to TGF-β. TGF-β CAR-T cells improve the antitumor efficacy of neighboring cytotoxic T cells and do not result in the preferential expansion of Treg cells nor do TGF-β CAR-Treg cells cause CAR-mediated suppression of Teff cells [163].

The interaction of CAR-T cells with immunoregulatory environments has been shown by CAR-T memory formation and function via hematopoietic stem cells (HSC) [98], which were shown to be common in multiple cancer types and appeared supported by CAF-originated cytokines [136]. Tumor-associated myelopoiesis was recently demonstrated, and this was shown to generate immune-suppressive myeloid populations [164].

CAF induce the differentiation of tumor-associated mast cells (MC), which are known to induce angiogenesis and modulate chronic inflammation [136] and potentially cancer cell growth [165].

The analysis of spatially-defined immune landscapes of triple-negative breast cancer (TNBC) showed distinct clinicopathologic features, RNA-based immune signatures, and spatially defined protein-based tumor–immune microenvironment in early-stage PD-L1+ and PD-L1−TNBC. PD-L1+ stromal and intraepithelial cells showed spatially specific alterations in CTLA-4, and spatially defined PD-L1+ cells were enriched in several clinically actionable immune proteins, with an impact on therapy efficacy and disease outcome. Gene expression signatures obtained from single-cell RNA sequencing have correspondingly been used to characterize CAR-T strategy outcomes, CAR design impact, and combination with ancillary therapies [89].

### 6.2. CAR-T Cell Interactions with Endogenous Immune Cell Subpopulations

Mentions of functional interaction of CAR-T cells with endogenous immune cell subpopulations are reported in the relevant subject sections above. However, a special mention is required here for experimental approaches that were explicitly designed to impact on such cell subpopulations.

In addition to their direct cytotoxic effect, allogeneic CAR invariant NKT (Natural Killer T)(CAR-iNKT) cells were found to exert potent antitumor effects through the cross-priming of CD8-T cells [166]. This resulted in a potent antitumor effect that lasted longer than the physical persistence of the allogeneic cells [166].

CAR-T engineered to secrete the combination of CCL19, and IL-7 were found to recruit endogenous immune cells and favor the generation of memory cells against cancer cells [26]. In a syngeneic, CD20-expressing mastocytoma mouse model, CCL19/IL-7 CAR-T cells induced the robust recruitment of endogenous T cells and DC, resulting in enhanced and durable tumor clearance [26]. IL-18 CAR-T cell treatment was accompanied by an overall change in the immune cell landscape associated with the tumor. More specifically, CD206− M1 macrophages and NKG2D+ NK cells increased in number, whereas Treg, suppressive CD103+ DC, and M2 macrophages decreased [28]. The antitumor activity of IL-12-secreting CAR-T cells was found to require both CD4+ and CD8+-T cell subsets, autocrine IL-12 stimulation, and subsequent IFNγ secretion by the CAR+-T cells [29].

The surface expression of 4-1BBL on CAR-T cells was proposed to remodel the TME through the autocrine-induced secretion of type I IFN, which improves DC cross-priming, Treg inhibition, and angiogenesis suppression [167].

CD40L expression on CD19 CAR-T cells resulted in the elevated surface expression of co-stimulatory molecules, adhesion molecules, HLA molecules, and the Fas death receptor on CD40+ tumor cells, thus increasing their immunogenicity [168]. These T cells also induced the secretion of pro-inflammatory IL-12 by DC in vitro and showed enhanced antitumor efficacy in vitro and in vivo [168]. The increased recruitment of macrophages, DC, and endogenous CD4+ and CD8+ T cells to lymphatic tissues was observed, along with the recruitment of DC, CD4+, and CD8+ T cells to the tumor. Endogenous T cells were activated to suppress antigen-negative tumor re-challenge, suggesting induced epitope spreading [169].

T cells engineered to secrete Fms-like tyrosine kinase 3 ligand (Flt3L), a hematopoietic cell growth factor, promote intratumoral DC1 and DC-precursor proliferation [170]. Furthermore, when T cells were co-transduced to express Flt3L and an anti-HER2 CAR, a combined treatment with these CAR-T cells and adjuvants induced an enhanced antitumor response and epitope spreading in endogenous T-cells [170].

Finally, CAR-T cells can be engineered to facilitate the engagement of tumor cells by endogenous, non-engineered T cells through the secretion of bispecific T-cell engagers (BiTE), which are composed of two fused scFvs [171]. An example is engineered BiTE with one scFv targeting EGFR, which is overexpressed in glioblastoma cells, and the other targeting CD3 on T cells. EGFRvIII-targeting CAR-T cells engineered to secrete EGFR/CD3 BiTE have been shown to eliminate orthotopic tumor xenografts with heterogeneous EGFRvIII expression [171].

## 7. Conclusions

Future research directions may benefit from designing anticancer CAR-T strategies according to the complexity of immune cells and stromal populations in the cancer milieu. The potential crosstalk among different engineered features within the CAR-T cells, as well as with endogenous immune cells, tumor cells, and TME, requires being taken into consideration for enhancing CAR-T cell therapy and reducing toxicity in the treatment of solid tumors.

To enhance synergy with endogenous T cell signaling appears to be an effective strategy for improving anticancer responses [172]. Treg can attenuate the cytolytic activity and proliferation of CD8+ T cells during adoptive immunotherapy [94] through the secretion of immunosuppressive interleukins and TGF-β and induced expression of the CTLA-4 and PD-1 checkpoint inhibitors. The depletion of Treg in a variety of tumor types was shown to trigger an antitumor immune response [21] and to induce CAR-T cell differentiation toward memory cells [99] for long-term immune recognition and anticancer responses [100]. Thus, lymphodepletion may be useful before CAR-T cell therapy, as it not only decreases the number of immunosuppressive Treg but also reduces competition from other lymphocyte populations [21].

The role of B cells in anticancer responses has only recently been explored. B cells largely reside in lymphoid aggregates. These were enriched in memory B cells in responder tumors, suggesting the course of an effective antitumor response. The mechanism underlying the effect of B cells in response to adoptive immunotherapy is poorly understood. B cells may activate T cells through antigen presentation by memory B cells and B cells-derived cytokines [129,130,131].

Inhibiting the activity of M2-like TAM and redirecting the polarization of macrophages toward the M1 phenotype appears to be a promising approach to enhance response to adoptive immunotherapy [21]. As type I DC can effectively stimulate CD8+ T cells in tumor-draining lymph node and within the tumor, efforts to increase these cells in the tumor microenvironment may provide an effective immunostimulatory pathway.

NK cells, neutrophils, and DC infiltrate the tumor microenvironment and further impact antitumor immune responses. The activation of NK cells leads to the recruitment and activation of other effectors cell subsets, such as dendritic cells, macrophages, and neutrophils, and to the subsequent activation of antigen-specific T and B cell responses. Tumor-infiltrated neutrophils have shown both pro- and antitumor phenotypes.

Corresponding conditioning regimens/ablation of immunosuppressive cell subpopulations and CAF subtypes before CAR-T cell infusion may pave the way for higher efficacy. The discovery of the tumor-restraining functions of CAF may provide a potential explanation for the unsuccessful clinical trials of therapeutic agents targeting CAF or stromal components. These observations suggest that future therapeutic strategies should avoid the generic targeting of tumor-restraining CAF subpopulations, rather favoring targeting tumor-promoting CAF subsets. FAP-expressing CAF can have an immunosuppressive function, which might present an opportunity for improving immunotherapies via the depletion of these CAF or targeting CAF-derived factors such as CXCL12 or CCL2.

Finally, state of the art approaches, such as single-cell sequencing, as well as proteomic and metabolomic analyses, are helping to dissect the heterogeneity and functional interactions of different cell types in the tumor microenvironment. This acquired knowledge may significantly enhance our ability to manage these complex interactions for the engineering of the next generations of CAR for solid tumor therapy [14,93,173].

## Figures and Tables

**Figure 1 ijms-23-00405-f001:**
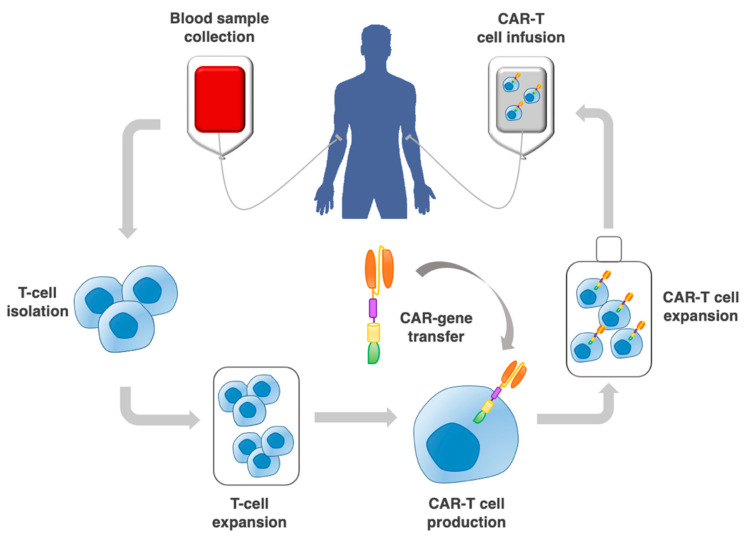
Procedure to implement adoptive CAR-T cell therapy. This includes the collection of a patient blood sample; T cell selection by leukapheresis from peripheral blood; CAR–gene transfer through a vector; expansion of CAR-T cells in vitro, and re-infusion into the patient.

**Figure 2 ijms-23-00405-f002:**
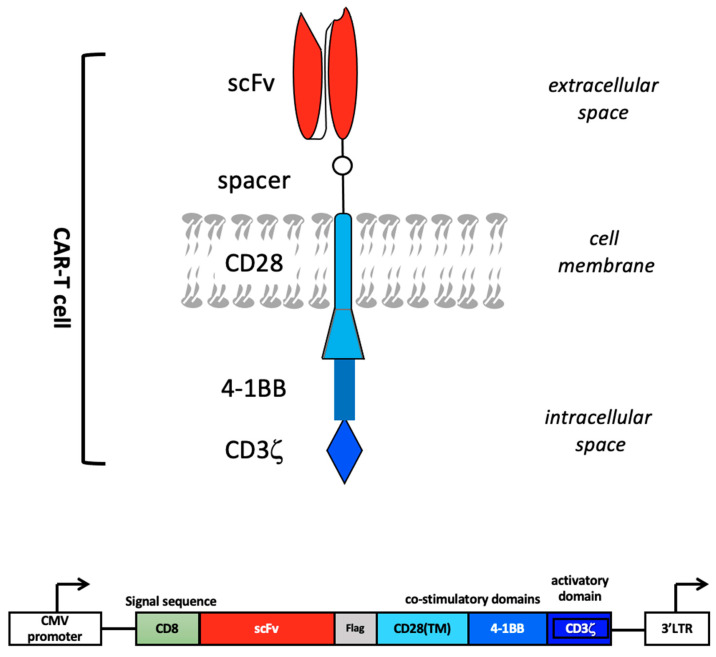
CAR design. The extracellular portion of the CAR molecule is typically derived from a monoclonal antibody that recognizes a cancer-associated antigen. The variable heavy (VH) and light (VL) chains, or single-chain variable fragment (scFv), are connected by a hinge to form the antigen-binding region of the CAR molecule. The antigen-binding region is linked through a transmembrane domain to an intracellular T-cell signaling domain, in particular CD3ζ, which is the primary activation domain for TCR mediated T-cell activation. A CAR construct further comprises one or more co-stimulatory domains, e.g., those derived from 4-1BB and CD28. Bottom: schematic representation of a CAR construct inserted in an expression vector, e.g., a lentiviral vector. Vector-derived CMV (cytomegalovirus)-promoter and 3′LTR (long terminal repeat) regions are depicted.

**Figure 3 ijms-23-00405-f003:**
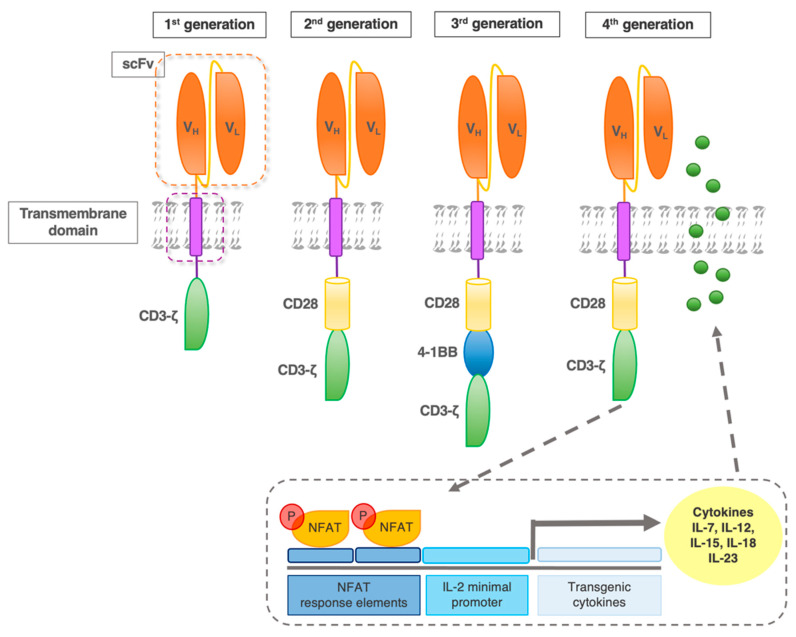
CAR evolution. First-generation CAR contains one intracellular signaling domain (CD3ζ) of the T cell receptor complex. Second-generation receptors contain one additional co-stimulatory domain (e.g., CD28). Based on the most effective second-generation CAR, the third generation added another co-stimulatory molecule, such as 4-1BB. Fourth-generation CAR were designed to activate NFAT (nuclear factor of activated T cells) transcription factor-driven cytokine production. After the CAR recognizes the target antigens, CD3ζ mediates downstream signaling and the activation/phosphorylation of NFAT. P-NFAT is shuttled into the nucleus and binds to the NFAT response elements/interleukin(IL)-2 minimal promoter of a transgenic expression cassette co-transfected with the CAR construct.

**Figure 4 ijms-23-00405-f004:**
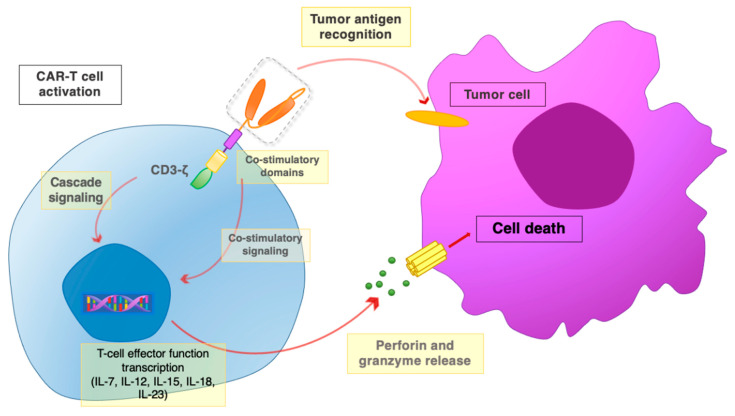
The CAR-T interaction with tumor cells. Once the scFv portion recognizes and binds the tumor antigen, the co-stimulatory domains and the CD3*ζ* chain promote activatory signaling cascades. Downstream signaling leads to the activation of T cell effector functions, with the release of perforin and granzyme, leading to the death of target tumor cells.

**Figure 5 ijms-23-00405-f005:**
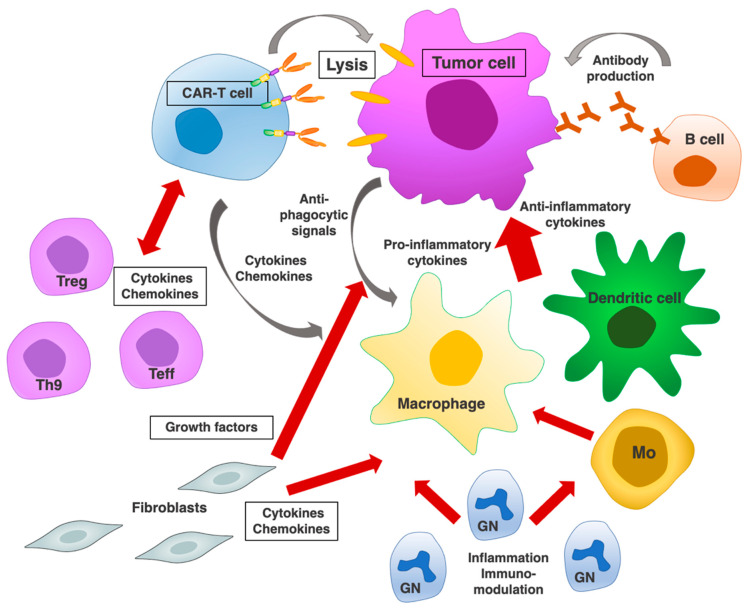
CAR-T cells can trigger endogenous tumor-specific immune response. CAR-T cell activation increases infiltration by dendritic cells, macrophages, and B cells in the target tumor. Additional infiltrating populations include granulocytes (GN), monocytes (Mo), T helper cells (Th9), T effectors (Teff), and T regulatory cells (Treg). Key interactions of infiltrating immune cell populations with cancer cells are mediated by soluble cytokines/chemokines, among them CCL/CCR, IL-2, IL-10, IL-35, TGF-β (Treg, Teff, B cells), CSF1/CSF1R, CCL2/CCR2, IL-10 (TAM), IL-2, IL-15, IL-18, IL-21 (NK), HGF, IGF1, CTGF, PDGF, VEGF, LIF, cytokines, chemokines, BMP4, TGF-β (CAF), and inflammatory mediators (TAM, GN, CAF) [101]. The cell–cell interactions are described in detail in each reference section.

## Data Availability

Not applicable.

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
