# Peer review of "Cancer-Homing CAR-T Cells and Endogenous Immune Population Dynamics"

_ijms, 2021, doi:10.3390/ijms23010405_

Round 1

Reviewer 1 Report

In this manuscript: Cancer-Homing CAR-T Cells and Endogenous Immune Population Dynamics

The authors want to give an overview of the CAR T cell therapy and its interaction with the responses mounted against the tumor.

In my opinion this manuscript lacks focus on CAR T cell therapy.

  1. Introduction

“NK cells are cytotoxic lymphocytes that identify infected cells in 43 the absence of MHC. The lack of endogenous antigen specificity for targeting cells to be”

The title of the paper is about CAR T cells, and CAR NK cells are not mentioned anymore, I don’t think this is needed

-“CAR-T cell therapy for solid tumors faces many hurdles [8, 9], starting from the very 59 first step of intravenous administration of activated CAR-T cells, .....

There are more hurdles as important as these that are not mentioned and are important for the scope of this manuscript such as infiltration and persistence

-2.2. The fourth-generation CAR

Is there already any benefit of this approach in patients? which type of solid tumors?

Immunomodulation of the TME is very important in solid tumors, affecting CAR T cell infiltration and persistence

I will appreciate more details in this part

-3. CAR-T against solid tumors

All the paper is about CAR T cells in solid tumors, I think a different title should be chosen

CAR T cell in solid tumors face different challenges that are expanded at this point

3.1 Effective CAR design....

This is a very broad title and you just talk about tumor antigens here.

What was the benefit in clinical trials of using those CAR T cells?

Also, you should mention CAR T cells targeting multiple antigens to avoid antigen scape, and add some references

3.2 Treatment toxicity

Also need to mention ICANS

-3.3 T cell exhaustion

The cellular product is important, which T cell subpopulation are using to manufacture CAR T cells,in the context of the paper

-4 Cancer-associated immune cells populations

all this part should be put in the context of CAR T cell therapy, in points 4.2, 4.3, 4.4 and 4.5 you don't mention how these cells are affecting CAR T cell therapy responses, you mention ICB, but that is not the point of the paper.

What are the interactions of TAMS with CAR T cells, what studies preclinical and clinical are already studied that, what are the outcomes?

how all these immune cells are interacting (intertwining) with CAR T cells in solid tumors? There is no mention of that at all

Is there any CAR T design that has been made taking these interactions into account to increase efficacy in solid tumors? Are they in clinical trials already?

Examples: antiFOLR2 CAR T cells targeting TAM in murine models of ovarian cancer, colon cancer, and melanoma 

TGFβresistant EGFRvIII CAR T showing increased survival

CAR T cells expressing IL-8 receptors (CXCR1 or CXCR2) 

.....

-5. Cancer-associated fibroblasts

Same, please put into  CAR T cells context and how are affecting this therapy and what has been done to overcome this challenge

In general, more references are needed all over the text and to put every point in the context of the interaction with CAR T cells,

Some tables may help

Author Response

In this manuscript: Cancer-Homing CAR-T Cells and Endogenous Immune Population Dynamics

The authors want to give an overview of the CAR T cell therapy and its interaction with the responses mounted against the tumor.

In my opinion this manuscript lacks focus on CAR T cell therapy.

- Thank you for your comments and detailed work you conducted on our text, as it allows to considerably improve our review. Extensive redrafting was conducted for a better focus on CAR-T therapy.

  1. Introduction

“NK cells are cytotoxic lymphocytes that identify infected cells in 43 the absence of MHC. The lack of endogenous antigen specificity for targeting cells to be”

The title of the paper is about CAR T cells, and CAR NK cells are not mentioned anymore, I don’t think this is needed

- We agree on a better focus on CAR-T cells. We have changed the text accordingly.

-“CAR-T cell therapy for solid tumors faces many hurdles [8, 9], starting from the very 59 first step of intravenous administration of activated CAR-T cells, .....

There are more hurdles as important as these that are not mentioned and are important for the scope of this manuscript such as infiltration and persistence

- Infiltration, persistence, exhaustion CAR-T cells and distinct interaction with other cancer-inflitrating cell populations are discussed all along the article. We have extended, though, your comment after this sentence.

-2.2. The fourth-generation CAR

Is there already any benefit of this approach in patients? which type of solid tumors?

- We have added updates on ongoing clinical trials, on fourth generation CARs.

Immunomodulation of the TME is very important in solid tumors, affecting CAR T cell infiltration and persistence

I will appreciate more details in this part

- we have extended comments on immunomodulation of TME. We have added details on the impact of immunomodulatory approaches on CAR-T cell function.

Knowledge on long-term activation of CAR-T cells has been recently gained. Less is known on drivers of tumor infiltration, despite considerable focus e.g. on the role of chemokines secretion on disease outcome according to chemokine production (Yong et al. Immunol Cell Biol, 2017). We have added relevant information on this topic.

-3. CAR-T against solid tumors

All the paper is about CAR T cells in solid tumors, I think a different title should be chosen

- thank you for your suggestion, we have changed the title accordingly.

CAR T cell in solid tumors face different challenges that are expanded at this point

  • Effective CAR ...

This is a very broad title and you just talk about tumor antigens here.

- thank you again for your suggestion, we have changed the title accordingly.

What was the benefit in clinical trials of using those CAR T cells?

Also, you should mention CAR T cells targeting multiple antigens to avoid antigen scape, and add some references

- we have added the requested information and references.

  • Treatment toxicity

Also need to mention ICANS

- we have added the requested reference to occurrence and management of ICANS.

-3.3 T cell exhaustion

The cellular product is important, which T cell subpopulation are using to manufacture CAR T cells,in the context of the paper

- thank you for your suggestion. We have added experimental examples relevant to this issue.

-4 Cancer-associated immune cells populations 

all this part should be put in the context of CAR T cell therapy, in points 4.2, 4.3, 4.4 and 4.5 you don't mention how these cells are affecting CAR T cell therapy responses,

- we have added more information on therapeutic impact of several of the discussed procedures in the related sections of the review.

you mention ICB, but that is not the point of the paper.

- we have better clarified our mentions of ICB, both to relieve T cell exhaustion and as a discovery tool for immunosuppressive mechanism in tumors.

What are the interactions of TAMS with CAR T cells, what studies preclinical and clinical are already studied that, what are the outcomes?

- we have added considerable more information on TAM, related intervention approaches and impact on cancer growth/disease outcome.

how all these immune cells are interacting (intertwining) with CAR T cells in solid tumors? There is no mention of that at all

- we respectfully disagree. Interactions of CAR-T cells with immune cells in solid tumors and functional outcomes were mentioned in paragraphs where relevant. We have now extended the information provided on this topic in a dedicated section.

Is there any CAR T design that has been made taking these interactions into account to increase efficacy in solid tumors? Are they in clinical trials already?

Examples: antiFOLR2 CAR T cells targeting TAM in murine models of ovarian cancer, colon cancer, and melanoma

TGFβresistant EGFRvIII CAR T showing increased survival CAR T cells expressing IL-8 receptors (CXCR1 or CXCR2)

- thank you for your suggestions, we have added relevant information and references, and revised our text accordingly.

-5. Cancer-associated fibroblasts

Same, please put into CAR T cells context and how are affecting this therapy and what has been done to overcome this challenge

- we have extensively revised and extended the CAF section and added the requested information and references.

In general, more references are needed all over the text and to put every point in the context of the interaction with CAR T cells,

Some tables may help

- thank you for pinpointing the issue. We have added focused references to most sections of the review and wherever experimental detail was needed.

Reviewer 2 Report

  1. A serious drawback of this review is the low quality of the references.When citing facts in the review, authors often refer not to original studies or reviews, but to other articles that are devoted to a different study, but which also cite similar information. For example, a similar picture can be found in lines 165, 166. It is highly advisable in such cases to provide reference to original research.
  2. Some parts of review contain very few references. For example, in part “ Cancer-associated fibroblasts” the same one references is used several times and only at its end there is a references to another article.
  3. In Part “3. CAR-T against solid tumors”, only a third of the sections (3.1) are devoted to solid tumors therapy by CAR-T.Sections 3.2. and 3.3 are devoted to general problems of CAR-T.
  4. Section 4 is loosely related to the main topic of the review.Various types of tumor immune cells are listed, but their contribution to the effectiveness of CAR-T or NK therapy is hardly discussed. The picture is similar in section 5. Such an analysis can be found in the Discussion section. Probably, the authors should transfer the relevant information about the role of certain cells on the effectiveness of therapy from the discussion section to sections 4 and 5, in accordance with the described cell types.
  5. Figure 5 is difficult for understanding. it is worth remaking this Figure or accompanying it with a detailed legend.

Author Response

Comments

and Suggestions for Authors

  1. A serious drawback of this review is the low quality of the references. When citing facts in the review, authors often refer not to original studies or reviews, but to other articles that are devoted to a different study, but which also cite similar information. For example, a similar picture can be found in lines 165, 166. It is highly advisable in such cases to provide 

    reference to original research.

  2. Some parts of review contain very few references. For example, in part “ Cancer-associated fibroblasts” the same one references is used several times and only at its end there is a references to another

-  thank you for pinpointing the issue and for all your comments, as they allowed to considerably improve our review. We have added focused references to most sections of the review and wherever experimental detail was needed.

  1. In Part “3. CAR-T against solid tumors”, only a third of the sections (3.1) are devoted to solid tumors therapy by CAR- T.Sections 3.2. and 3.3 are devoted to general problems of CAR-T

-  thank you for pinpointing the issue. Solid tumor therapy has been extended in additional sections of the review. We have added focused references to the main focus of the review, i.e. the added efficacy or a worse outcomes according to distinct interactions with distinct immune cell populations and related soluble factors in the tumor microenvironment.

  1. Section 4 is loosely related to the main topic of the review.Various types of tumor immune cells are listed, but their contribution to the effectiveness of CAR-T or NK therapy is hardly discussed. The picture is similar in section Such an analysis can be found in the Discussion section. Probably, the authors should transfer the relevant information about the role of certain cells on the effectiveness of therapy from the discussion section to sections 4 and 5, in accordance with the described cell types.

–  thank you for your suggestion. We have extensively re-drafted, and considerably extended, distinct sections of the review accordingly.

  1. Figure 5 is difficult for it is worth remaking this Figure or accompanying it with a detailed legend.

–  we have extended the legend to Figure 5 accordingly.

Round 2

Reviewer 2 Report

The review has been largely revised. In the current version, the discussion section seems redundant and may be deleted. 

Author Response

thank you for your comment. We have eliminated the Discussion section, and selectively strengthened the Conclusions section.